# Trop-2 as a Therapeutic Target in Breast Cancer

**DOI:** 10.3390/cancers14235936

**Published:** 2022-11-30

**Authors:** Elizabeth Sakach, Ruth Sacks, Kevin Kalinsky

**Affiliations:** Winship Cancer Institute, Emory University, Atlanta, GA 30322, USA

**Keywords:** Trop-2, breast cancer, antibody drug conjugate

## Abstract

**Simple Summary:**

Trop-2 is an exciting, new target for the treatment of breast cancer. Trop-2 is found at high levels in multiple cancers such as prostate, pancreatic, urothelial, lung, and breast cancer. Among different breast cancer subtypes, Trop-2 is most highly expressed in triple negative breast cancer. Drugs that inhibit Trop-2 are now an important treatment option for patients with metastatic triple negative breast cancer, for whom few treatment options exist. The benefit of Trop-2 inhibitors has also been observed in patients with hormone receptor positive breast cancer, whose tumors are resistant to standard treatments. Ongoing studies are working to understand if patients can benefit from different drug combinations with Trop-2 inhibitors in the metastatic setting and if Trop-2 inhibition can benefit patients with early stage disease.

**Abstract:**

The emergence of Trop-2 as a therapeutic target has given rise to new treatment paradigms for the treatment of patients with advanced and metastatic breast cancer. Trop-2 is most highly expressed in triple negative breast cancer (TNBC), but the receptor is found across all breast cancer subtypes. With sacituzumab govitecan, the first FDA-approved, Trop-2 inhibitor, providing a survival benefit in patients with both metastatic TNBC and hormone receptor positive breast cancer, additional Trop-2 directed therapies are under investigation. Ongoing studies of combination regimens with immunotherapy, PARP inhibitors, and other targeted agents aim to further harness the effect of Trop-2 inhibition. Current investigations are also underway in the neoadjuvant and adjuvant setting to evaluate the therapeutic benefit of Trop-2 inhibition in patients with early stage disease. This review highlights the significant impact the discovery Trop-2 has had on our patients with heavily pretreated breast cancer, for whom few treatment options exist, and the future direction of novel Trop-2 targeted therapies.

## 1. Introduction

Human trophoblastic cell surface antigen 2 (Trop-2) is a transmembrane calcium signal transducer highly expressed in multiple tumor types on the membrane surface of epithelial cells. Under physiologic conditions, Trop-2 plays a critical role in embryonic development, placental tissue formation, and stem cell proliferation. Trop-2 is expressed at low levels on the surface of many types of normal epithelial cells such the heart, liver, kidney, and lung. Expression on these normal epithelial cells is at a much lower level than in epithelial tumors, making Trop-2 an ideal therapeutic target [1,2,3]. When Trop-2 is overexpressed, it acts as an oncogene promoting tumor proliferation, growth, invasion, and metastasis in epithelial cancers such as breast, colon, prostate, pancreatic, urothelial, and lung [4,5,6,7,8,9,10,11]. High Trop-2 expression is associated with a poor prognosis in multiple cancer types, with worse overall survival and disease-free survival outcomes [12,13,14,15,16].

## 2. Trop-2 Expression in Breast Cancer 

Trop-2 is overexpressed in all breast cancer subtypes, however it is most elevated in triple negative breast cancer (TNBC) as compared to estrogen receptor positive (ER+) or HER-2+ tumors [2,16]. Aslan et al. evaluated Trop-2 protein levels in breast cancer tumors by immunohistochemistry (IHC). High levels of Trop-2 expression were found in 50% of ER+ (*n* = 22), 74% of HER2+ (*n* = 35), and 93% of TNBC samples (*n* = 28) [16].

Vidula et al. 2017, studied the associations of Trop-2 expression with clinical characteristics and outcomes from microarray data from neoadjuvant I-SPY1, METABRIC, and TCGA patient databases. In all 3 datasets, Trop-2 had a wide range of expression in all breast cancer subtypes, particularly luminal A and TNBC. Presence of Trop-2 was associated with expression of genes central for cell epithelial transformation, adhesion, and proliferation. Trop-2 expression was inversely related to the expression of immune genes, potentially affecting tumor growth. These findings supported the ability of Trop-2 to be used as a therapeutic target across a variety of breast cancer subtypes [17].

### 2.1. Trop-2 as an Oncogene in Breast Cancer 

Trop-2 is a critical element in TNBC tumor growth. Trop-2 gene deletion and gene silencing has been found to suppress TNBC cell growth in vitro and in vivo. Aslan et al. studied Trop-2 gene deletion via CRISPR/Cas9 technology in TNBC cells that endogenously express Trop-2. The loss of Trop-2 significantly suppressed TNBC cell growth in colony formation and proliferation assays. Using small hairpin RNA to create Trop-2 knockdown cells, downregulation of Trop-2 also significantly impaired the colony-forming ability and proliferation of TNBC cell lines. Downregulation of Trop-2 dramatically decreased the invasion ability of the TNBC cell lines in three-dimensional Matrigel drop invasion assays [16].

In preclinical models, Aslan et al. subcutaneously implanted Trop-2 depleted TNBC cells into mice. Tumor volumes were measured every three days. With the downregulation of Trop-2 in the implanted TNBC mouse tumors, there was a significant delay in tumor growth and decrease in tumor weight. The Trop-2 gene knockout models demonstrated the oncogenic potential of Trop-2 expression in TNBC [16].

Further studies by Aslan et al. suggest Trop-2 may lead to an oncogene-mediated metabolic reprogramming in TNBC by regulating a group of metabolic genes and oncogenes. The investigators evaluated changes in protein levels in TNBC tumors upon modulation of Trop-2. Trop-2 expression was found to be related to increased levels of a 5-gene metabolic signature (comprising of TALDO1, GPI, LDHA, SHMT2, and ADK). This 5-gene metabolic signature was associated with oncogenic metabolism and poorer overall survival in early stage breast cancers. Aslan et al. found this data was clinically correlated, as patients with the 5-gene metabolic signature had worse overall survival and disease-free survival in 12 different mRNA expression datasets of breast cancer patients [16].

As an oncogene, Trop-2 plays a role in several major signaling pathways involved in cell proliferation, but its precise role in these pathways is not completely understood. Trop-2 acts as a calcium signal transducer, leading to activation of various tumorigenic pathways including NF-KB, cyclin D1, and ERK [18,19,20,21,22]. Through calcium-mediated signal cascades, Trop-2 activates the ERK1/2-MAPK pathways, which modulate cell cycle progression and promote the evasion of apoptosis [22]. The Bcl-2 family is a crucial checkpoint in apoptosis, comprising anti-apoptotic proteins Bcl-2, Bcl-xl, and Mcl1 and pro-apoptotic proteins, Bax, Bak, and Bim [23]. The expression of Bcl-2 and Bax can be regulated by the MEK/ERK pathway [24,25].

Lin et al. proposed that a fragment antigen-binding fragment (Fab) against Trop-2 could inhibit the evasion of pro-apoptotic pathways and potentially induce apoptosis to optimize responses to anti-cancer therapeutics. The group employed a human Fab phage library to isolate a human Fab that recognized the extracellular domain of Trop-2 [26]. Specific binding of Trop-2 Fab to Trop-2 on the surface of breast cancer cells was confirmed by ELISA, flow cytometry, and fluorescent staining. Investigators performed an MTT assay, a colorimetric assay to assess metabolic activity, and showed that Trop-2 Fab was effective at inhibiting proliferation in the TNBC cell line MDA-MB-in a dose-dependent manner. Immunofluorescence staining and Western blot analysis showed Trop-2 Fab upregulated Bax expression, a pro-apoptotic protein, and downregulated Bcl-2 expression, an anti-apoptotic protein, suggesting a Trop-2 Fab could induce apoptosis in TNBC cells. 

In vivo, the researchers confirmed an antitumor effect of the Trop-2 Fab in a breast cancer xenograft model. The tumor inhibition rate was 28% in the high dose Trop-2 Fab (30 mg/kg) treated group. Consistent with in vitro studies, IHC and Western blot analysis of excised TNBC tumors from mice showed significantly downregulated Bcl-2 expression and upregulated Bax expression compared with the control group in treated Trop-2 Fab mice. The group found that high concentrations of Trop-2 Fab allowed for significant tumor inhibition, confirming the hypothesis that Trop-2 could be a candidate for the therapeutic inhibition of TNBC [26].

### 2.2. Trop-2 as Determinant for Breast Cancer Survival 

Trop-2 promotes migration of invasive breast cancer cells by inducing the epithelial-mesenchymal transition (EMT) which contributes to metastasis. Increased Trop-2 expression is associated with lymph node involvement and distant metastasis [27]. In a study evaluating Trop-2 expression in 127 patients with early stage TNBC, patients with high Trop-2 expression as determined by IHC, had higher rates of nodal involvement (53%) compared to patients with medium (23%) and low (21%) Trop-2 expression (*p* = 0.03). Lymphovascular invasion was also more frequent and found in 65% of patients with high Trop-2 versus 15-16% of patients with medium and low Trop-2 expression, respectively (*p* ≤ 0.001) [28].

Trop-2 is synthesized in the endoplasmic reticulum, glycosylated in the Golgi apparatus, and sorted to the cell membrane. Trop-2 can then be activated by antibody-mediated cross-linking of cell-surface molecules or cleaved in the intramembrane. Some Trop-2 is also retained in intracellular compartments at different levels in various tumors. Ambrogi et al. found that the activation state of Trop-2 is a critical determinant of breast cancer tumor progression and could implicate Trop-2 as a novel prognostic indicator in breast cancer. Observations by this group found that Trop-2 localization and glycosylation are associated with worse overall survival, whereas intracellular retention is associated with less frequent disease relapse and therefore better overall survival [29]. By localizing Trop-2 expression via IHC in 702 breast tumor samples from patients, Ambrogi et al., found that localization of Trop-2 in the membrane is an unfavorable prognostic factor for overall survival (HR 1.63; *p* = 0.04), whereas intracellular Trop-2 had a favorable impact on prognosis and disease relapse (HR 0.48; *p* = 0.003) [29].

Lin et al. examined a panel of invasive ductal carcinoma (IDC) tissues (*n* = 82) and adjacent non-malignant tissue controls (*n* = 70) from patients undergoing surgery at a single institution [30]. None of the patients had received neoadjuvant chemotherapy or immunotherapy. The investigators evaluated Trop-2 expression by IHC and found significantly higher Trop-2 expression in IDC tissue compared to normal controls. High expression of Trop-2 was related to increased histologic grade (*p* = 0.002), P53 status (*p* = 0.004), cyclin D1 status (*p* < 0.01), lymph node metastasis (*p* < 0.01), distant metastasis *p* = 0.004) and more advanced TNM staging (*p* < 0.01). In contrast, no statistically significant association was found between Trop-2 expression and age at diagnosis, tumor size, hormone receptor status, HER-2 status, or Ki-67. The strongest predictors of poor survival were high Trop-2 expression (*p* = 0.03), high cyclin D1 expression (*p* = 0.04), and lymph node metastasis (*p* = 0.06) [30]. Trop-2 is associated with high cyclin D1 expression as Trop-2 must bind to cyclin D1 to become an oncogene [31]. The binding of the two molecules affects the stability of cyclin D1 and increases cell proliferation and survival [1]. Silencing of this fusion protein has been shown to inhibit tumor growth [32]. 

## 3. Trop-2 as a Therapeutic Target: From Bench to Bedside

Precision medicine has opened the door to new possibilities in the world of targeted therapy development. Given the unmet need for innovative, new therapies in the treatment of TNBC, the emergence of Trop-2 is an exciting path forward as Trop-2 expression is reported in ~85% in TNBC tumors [16,33]. The first FDA (Food and Drug Administration) approved Trop-2 inhibitor for the treatment of TNBC is an antibody-drug conjugate (ADC). Over the past two decades, ADCs have moved to the forefront of cancer care and precision medicine. By targeting specific antigens on tumor cells, ADCs have become a leading therapeutic in Trop-2 inhibition [34].

## 4. Anti-Trop-2 Antibody Drug Conjugates: An Exciting Path Forward

ADCs are comprised of a monoclonal antibody, a cytotoxic agent (payload), and a linker that connects the monoclonal antibody to the cytotoxin. For Trop-2 directed ADCs: (1) the monoclonal antibody recognizes and binds to Trop-2 on the tumor cell; (2) the payload is internalized into the tumor cell; (3) the payload undergoes intracellular trafficking that carries it to the lysosomes; (4) following antibody catabolism and hydrolysis of the linker, the payload is released and induces apoptotic cell death [34,35,36]. Neighboring cancer cells (even with no target antigen present) are impacted by the bystander effect, which occurs when the cytotoxic payload is released from the target cell or within the extracellular space contributing to an augmented anti-tumor effect [37,38]. 

ADCs are a rapidly expanding class of agents with at least 160 drugs in the preclinical and clinical space [39,40]. Three ADCs are FDA approved for breast cancer treatment, but only one of these targets Trop-2. Extensive efforts are being made in ongoing trials to harness the effect of Trop-2 inhibition through ADC therapeutics. 

### 4.1. Sacituzumab Govitecan: The First FDA Approved Anti-Trop-2 ADC

In April 2020, the FDA granted accelerated approval to sacituzumab govitecan-hziy (Trodelvy), a Trop-2 directed ADC, for patients with unresectable locally advanced and metastatic TNBC who received two or more prior lines of therapy for metastatic disease. Sacituzumab govitecan is composed of a humanized anti-Trop-2 immunoglobulin (IgG) antibody conjugated through a hydrolysable linker to SN-38, the cytotoxic agent or payload. SN-38 is a topoisomerase I inhibitor and active metabolite of irinotecan, which induces double-stranded breaks in DNA [41,42]. Irinotecan, the prodrug of SN-38, has limited delivery of the active metabolite, SN-38, due to the 100 to 1000-fold higher potency it carries in comparison to irinotecan. This high potency causes toxicity and poor tolerability with 1/3 of patients experiencing grade 3 or 4 diarrhea [43]. Sacituzumab govitecan can deliver higher levels of SN-38 with an improved tolerance profile [44].

#### 4.1.1. Sacituzumab Govitecan in TNBC

The first-in-human study of sacituzumab govitecan originated as a basket trial which included 25 patients with 10 different epithelial cancers, including metastatic TNBC (*n* = 4), whose tumors had progressed on conventional treatments. Three of 25 patients had a >30% reduction in their tumor size, and one of these significant responders was a patient with metastatic TNBC [45]. A review of the initial data indicated improved response rates and clinical benefit in patients with TNBC given the starting dose of 10 mg/kg. Neutropenia was the only dose-limiting toxicity and 1 patient at this dose level experienced grade 3 diarrhea [46].

The efficacy of sacituzumab govitecan in TNBC was further evidenced in the multicenter, phase I/II, single-arm trial IMMU-132-01 that evaluated patients with advanced solid tumors, including 108 patients with metastatic TNBC who had received at least 2 therapies for metastatic disease [47]. This heavily pretreated population of metastatic TNBC demonstrated an overall response rate (ORR) of 33.3% (95% CI 24.6–43.1) and median response duration of 7.7 months (95% CI 4.9–10.8). The most common adverse events were nausea (67%), diarrhea (62%), fatigue (55%), and myelosuppression (74%) along with 9.3% of patients experiencing neutropenic fever [47,48]. Based on the findings from this trial, sacituzumab govitecan received accelerated approval by the FDA for the treatment of patients with metastatic TNBC. 

In the randomized phase III ASCENT trial, the use of sacituzumab govitecan was compared against chemotherapy of physician’s choice (eribulin, vinorelbine, capecitabine, or gemcitabine) in the treatment of 468 patients with relapsed or refractory metastatic TNBC. The median age of participants was 54 years, all patients had previous taxane exposure. The trial achieved its primary endpoint with a mPFS of 5.6 months with sacituzumab govitecan and 1.7 months with chemotherapy (HR for disease progression or death, 0.41; *p* < 0.001). The mOS was 12.1 months with sacituzumab govitecan and 6.7 months with chemotherapy (HR for death, 0.48; *p* < 0.001). The percentage of patients with an objective response was 35% with sacituzumab govitecan and 5% with chemotherapy (Table 1). Myelosuppression and diarrhea were more frequent with sacituzumab govitecan than with chemotherapy. Grade 3 or higher neutropenia was reported in 51% of patients treated with sacituzumab govitecan and 33% with chemotherapy, leukopenia in 10% and 5%, anemia in 8% and 5%, and diarrhea in 10% and <1% of patients, respectively [49].

An exploratory analysis was subsequently performed that assessed the potential clinical utility of Trop-2 expression. Patients with high, medium, and low Trop-2 expression that received sacituzumab govitecan demonstrated greater overall response rates (44%, 38%, 22%, respectively) compared to physician’s choice chemotherapy (1%, 11%, 6%). The majority of tumors had high expression of Trop-2 (*n* = 85; 56%), with a small proportion of tumors with low Trop-2 expression (*n* = 27; 18%) [50]. Regardless of Trop-2 expression, however, all patients with metastatic TNBC benefited from sacituzumab govitecan in comparison to physician’s choice chemotherapy. Trop-2 expression is not currently recommended to be checked as a biomarker to predict a benefit to sacituzumab govitecan. 

#### 4.1.2. Sacituzumab Govitecan in Hormone Receptor Positive Breast Cancer 

Given Trop-2 expression is also observed in hormone receptor positive subtypes, sacituzumab govitecan was evaluated for the treatment of HR+ (hormone receptor positive), HER2- metastatic breast cancers. As patients with HR+/HER2- develop tumor progression on endocrine therapy and targeted agents, this subtype remains challenging to treat, as options are limited to sequential single agent chemotherapy, which has lower response rates and greater toxicity. The HR+/HER2- cohort in IMMU-132-01 showed promising results in 54 patients who had received at least 1 line of hormone-based therapy and at least 1 prior chemotherapy in the metastatic setting. Overall response rates were 31.5% (95% CI 19.5–45.6%) and median PFS was 5.5 months (95% CI 3.6–7.6) [51].

In the randomized phase III TROPiCS-02 study, the use of sacituzumab govitecan was further evaluated in an endocrine resistant HR+/HER2- population whose tumors had progressed on at least 2–4 lines in the metastatic setting, with a median of 3 prior chemotherapies. Sacituzumab govitecan demonstrated an improvement in ORR (57% vs. 38%) and PFS (5.5 vs. 4.0 months; HR 0.66; *p* = 0.0003) compared to physician’s choice of chemotherapy (eribulin, vinorelbine, capecitabine, or gemcitabine) (Table 1) [52]. Overall survival data was presented at the European Society of Medical Oncology (ESMO) 2022 meeting with a median follow-up of 12.5 months, with 390 OS events. Sacituzumab govitecan improved mOS compared to physician’s choice chemotherapy (14.4 vs. 11.2 months; HR 0.79, *p* = 0.020) [53].

With the new approval of trastuzumab deruxtecan for patients with HER2-low breast cancers (low expression defined as a score of 1+ or 2+ on IHC with negative FISH), there are a proportion of patients who may receive trastuzumab deruxtecan and sacituzumab govitecan sequentially in the triple negative or HR+ setting if their tumors carry low HER2 expression [54]. There are currently no data on the effect of sequential ADC therapies and whether subsequent use of ADCs is an effective practice. 

### 4.2. Daptopotamab Deruxtecan 

The safety and efficacy of a new Trop-2 ADC, daptopotamab deruxtecan (dato-Dxd), is being investigated in the phase I study TROPION-PanTUMOR01 among patients with advanced solid tumors, including relapsed/refractory TNBC following standard of care treatment [55]. Dato-Dxd is a topoisomerase I inhibitor (exatecan derivative) attached to a humanized immunoglobulin G (IgG1) monoclonal antibody via a cleavable peptide linker and cysteine conjugation using thioether bonds [56,57]. Patients with TNBC enrolled on TROPION-PanTUMOR01, had received a median of 3 prior therapies (range 1–10) with 91% of patients having received a prior taxane and 30% a prior topoisomerase I inhibitor-based ADC. The interim data from the cohort of 44 patients with advanced or metastatic TNBC demonstrated an overall response rate by blinded independent central review (BICR) of 34% at a median follow up of 7.6 months and a disease control rate of 77%. Patients who had not received prior treatment with a topoisomerase I inhibitor-based ADC had better overall response rates (52%) and disease control rates (81%). The most common adverse events were nausea (58%), stomatitis (53%), alopecia (35%), vomiting (35%), and fatigue (33%) [55]. There is an ongoing dose expansion cohort in patients with HR+/HER2- breast cancer and the data will be presented at the San Antonio Breast Cancer Symposium in 2023 (NCT03401385). 

Two separate phase III TROPION-breast trials are now ongoing with dato-DXd versus investigator’s choice of chemotherapy in patients with metastatic TNBC who are not a candidate for a PD-L1 inhibitor (TROPION-Breast02; NCT05374512) and patients with metastatic HR+/HER2- breast cancer who have received at least one or two prior lines of systemic chemotherapy (TROPION-Breast01; NCT05104866). 

## 5. Ongoing Clinical Trials Involving Trop-2 Inhibition 

There are multiple promising clinical trials involving Trop-2 inhibition that are actively recruiting patients (summarized in Table 2, Table 3, Table 4 and Table 5). Many of these trials involve combining Trop-2 ADCs with immunotherapy (NCT04468061; NCT05382286; NCT04448886; NCT04434040; NCT04958785; NCT03971409; NCT03424005), PARP inhibitors (NCT04039230), or targeted therapy (NCT05143229) (Table 4). There are also ongoing investigations of sacituzumab govitecan and dato-DXd in the early stage and neoadjuvant space (NCT04595565, NCT04230109, NCT01042379, NCT01042379) (Table 5). 

### 5.1. Trials in the Metastatic Setting

#### 5.1.1. Patients with CNS Disease

There is an ongoing phase II trial sponsored by Southwest Oncology Group (SWOG) studying the effect of sacituzumab govitecan in patients with HER2-negative breast cancer with brain metastases. For this trial, participants must have experienced CNS progression after previous CNS-directed therapy (radiation, surgery, or any combination of therapy) with at least one measurable brain metastasis >1.0 cm in size, that has not been irradiated, or has progressed despite prior radiation therapy. The primary objective of the study is intracranial objective response rate and the secondary objectives are PFS, OS, and safety and tolerability of sacituzumab govitecan in this population. The investigators will also be evaluating ORR by hormone-receptor subtype (NCT04647916). 

#### 5.1.2. SKB264-01: A Novel Anti-Trop-2 ADC 

The SKB264-01 trial is a global, open label, multicenter phase I/II study of the novel anti-Trop-2 ADC SKB264 which delivers a belotecan-derived payload. Participants must have locally advanced or metastatic TNBC. Participants will also be enrolled with other solid tumors: ovarian, gastric, pancreatic, and bladder cancers. The primary endpoint for the phase II portion is ORR. Trop-2 expression will be evaluated for each participant retrospectively in archival tissue by IHC (NCT04152499) [58].

### 5.2. Combination Therapy with ADCs in the Metastatic Setting

#### 5.2.1. Immunotherapy + Anti-Trop-2 ADCs

Sacituzumab Govitecan with Immunotherapy.

The effect of immunotherapy with ADCs is not yet known, therefore studies are ongoing to evaluate if the addition of a programmed cell death ligand 1 (PDL-1) inhibitor will enhance the treatment response to Trop-2 inhibiting ADCs in patients with TNBC. The combination of sacituzumab govitecan plus pembrolizumab is currently being evaluated in the randomized, phase III ASCENT-04 trial versus treatment of physician’s choice (paclitaxel, nab-paclitaxel, gemcitabine, carboplatin) and pembrolizumab in participants whose tumors express PD-L1. The primary outcome is PFS (NCT05382286).

Alternatively, the combination of sacituzumab govitecan with pembrolizumab is being evaluated in a PD-L1 negative TNBC population in the phase II Saci-IO trial. The primary outcome for this study is PFS with secondary outcomes of OS, ORR, DOR, time to ORR, time to progression, and clinical benefit rate (NCT04468061). Additionally, the InCITe phase II trial is enrolling patients with metastatic TNBC to evaluate the combination of sacituzumab govitecan plus avelumab with the primary endpoint of ORR (NCT03971409) [59]. The combination of sacituzumab govitecan plus atezolizumab is also being evaluated in the phase I/II randomized umbrella study in patients with metastatic TNBC (NCT03424005).

A phase II study is being conducted combining sacituzumab govitecan with magrolimab in patients with metastatic TNBC. Magroimab is an anti-CD47 IgG antibody, which targets the macrophage immune checkpoint system rather than the T-cell checkpoint system as in PDL-1 therapy. Magrolimab inhibits the “don’t eat me signal” of cancer cells, allowing macrophages to engage their “eat me” function [60]. In cohort one of this trial, magrolimab will be combined with chemotherapy whereas in cohort 2, the drug will be combined with sacituzumab govitecan to determine a recommended phase II dose and evaluate whether the combination enhances the immunomodulatory effect of sacituzumab govitecan (NCT04958785). 

In the hormone receptor positive setting, the Saci-IO HR+ trial is enrolling patients in a phase II study of sacituzumab govitecan with or without pembrolizumab in patients with metastatic HR+/HER2- breast cancer (NCT04448886). 

Dato-DXd plus immunotherapy.

Preliminary results from the basket BEGONIA trial demonstrated positive activity and a good safety profile with the utilization of dato-DXd in metastatic TNBC. This two-part, open-label study was designed to determine the efficacy and safety of durvalumab in combination with novel oncology therapies (capivasertib, oleclumab, T-DXd, dato-DXd) with or without paclitaxel in addition to durvalumab and paclitaxel as first line treatment in metastatic TNBC. The study demonstrated improved response rates with the combination of durvalumab with dato-DXd (66.7%) compared to durvalumab and paclitaxel (58.3%). Most patients (73%) had PD-L1 low tumors (<5%) but responses were reported irrespective of PD-L1 status [61]. Due to these findings, dato-DXd was graduated into the expansion phase (NCT03742102). 

#### 5.2.2. PARP Inhibitors + Anti Trop-2 ADCs

The combination of sacituzumab govitecan with PARP inhibition could have a synergistic effect, given both drugs interrupt DNA replication and DNA repair. As the payload of sacituzumab govitecan is the active metabolite of irinotecan (SN-38), a topoisomerase I inhibitor, and PARP inhibitors block topoisomerase I cleavage complexes induced by topoisomerase I inhibitors, the DNA damage abilities are pronounced when used in combination, which unfortunately leads to dose-limiting myelosuppression when these therapies are given in combination. Bardia et al, evaluated the combination of sacituzumab govitecan with talazoparib in pre-clinical models and a phase 1b clinical trial (NCT04039230) for patients with metastatic TNBC. A staggered schedule with supportive therapy was relatively well-tolerated without DLTs, as predicted by preclinical models. The staggered schedule also demonstrated promising clinical activity with objective responses among patients with metastatic TNBC [62]. The trial has proceeded to phase II with the primary objective of evaluating dose-limiting toxicities and the secondary objectives are time to tumor response, DoR, PFS, and OS (NCT04039230). 

#### 5.2.3. Targeted Therapy + Anti-Trop-2 ADCs

The phase I ASSET trial is enrolling patients with metastatic TNBC to evaluate the combination of sacituzumab govitecan plus alpelisib, a PI3K inhibitor, initially approved in combination with fulvestrant for use in patients with metastatic HR+ HER2- breast cancer, based on the results of the SOLAR-1 trial. The investigators hypothesize the combination will have a synergistic effect as 8–25% of patients with TNBC have PI3K-activating mutations [63,64,65]. The primary outcome is to establish the recommended phase II dose of the combination and secondary outcomes are to measure the pharmacokinetics and overall response rate (NCT05143229). 

### 5.3. Trop-2 Inhibition in the Early Stage Setting

#### 5.3.1. Neoadjuvant Therapy with Sacituzumab Govitecan

The NeoSTAR trial is a phase II study enrolling participants to evaluate the efficacy of sacituzumab govitecan in the neoadjuvant TNBC setting in patients with at least one lesion that is >1 cm or greater in size with the absence of distant metastatic disease. After 4 cycles of sacituzumab govitecan, patients with biopsy-proven residual disease, have the option to receive additional neoadjuant therapy at the discretion of the treating physician. Initial results from 50 patients were presented at the 2022 American Society of Clinical Oncology (ASCO) national meeting. The majority of patients (98%; *n* = 49) completed 4 cycles of sacituzumab govitecan. Radiological response with sacituzumab govitecan alone was 62% (*n* = 31, 95% CI 48%, 77%). 26 of 49 patients proceeded directly to surgery after sacituzumab govitecan. The pathologic complete response (pCR) rate with sacituzumab govitecan alone was 30% (*n* = 15/50, 95% CI 18%, 45%) (NCT04230109) [66].

#### 5.3.2. Trop-2 Inhibition with Residual Disease after Neoadjuvant Chemotherapy 

The phase III SASCIA trial is currently investigating whether sacituzumab govitecan may be effective in patients with HER2- breast cancer who have residual disease after standard neoadjuvant chemotherapy (NACT). Patients enrolled on the trial must be at high risk of recurrence after at least 16 weeks of NACT (with at least 6 weeks of taxane therapy). Patients can have hormone receptor negative (<1% positive) cancer with any residual disease >ypT1mi or HR+ disease with clinical and post-treatment pathologic stage (CPS) and estrogen receptor status and grade (EG) score >3 or CPS + EG score 2 and ypN+ based on core biopsies before NACT. The primary endpoint of the trial is invasive disease free survival (iDFS). Secondary endpoints are OS and distant disease-free survival (NCT04595565). 

Results of a pre-planned safety interim analysis were presented at ESMO Breast Cancer 2022 and show the safety profile of sacituzumab govitecan was manageable in this setting. Sacituzumab govitecan was associated with a higher incidence of adverse events and dose delays than treatment of physician’s choice, but dose reductions occurred in similar frequency in both arms. Among the 88 patients with HER2-negative breast cancer at high risk of relapse after NACT, grade 1-4 adverse events (AEs) occurred in all 45 (100%) patients receiving sacituzumab govitecan and in 37 of 43 (86%) of patients with treatment of physician’s choice: capecitabine (*n* = 32) or observation (*n* = 11). Grade 3–4 AEs were 66.7% and 20.9%, respectively for each arm. Hematologic AEs along with all-grade nausea, vomiting, constipation, diarrhea, and alopecia were significantly more common with sacituzumab govitecan than physician’s choice. Dose delays were more frequent with sacituzumab govitecan, with at least one dose delay in 66.7% vs. 43.3% in patients receiving capecitabine. At least one dose reduction was required by 26.7% of patients receiving sacituzumab govitecan vs. 28.1% of patients receiving capecitabine [67]. Despite the increased toxicity associated with sacituzumab govitecan, the Trop-2 targeted ADC may be a good option for those patients with high-risk, residual disease, however the choice between capecitabine or sacituzumab govitecan in the adjuvant setting will need to be balanced with efficacy data from adjuvant pembrolizumab in TNBC patients who have residual disease. The dilemma regarding which adjuvant therapy is best for patients with residual disease will be in part addressed by results from the ASPRIA trial.

#### 5.3.3. Trop-2 Inhibition with Immunotherapy for Residual Disease 

In the phase II ASPRIA trial, investigators are evaluating the efficacy of sacituzumab govitecan in combination with atezolizumab, the anti-PD-L1 antibody, in patients with residual disease in the breast or lymph nodes after neoadjuvant chemotherapy. Participants may also enroll if they have circulating tumor DNA in the blood. The primary outcome is the rate of undetectable circulating tumor (cfDNA) after 6 cycles, with secondary outcomes of rate of undetectable cfDNA after 1 and 3 cycles, invasive disease-free survival rate, distant metastasis free survival rate, and OS at 3 years (NCT04434040). 

## 6. Promising Anti-Trop-2 Therapeutics in the Pre-Clinical Pipeline

### 6.1. TrMab-6: A Novel Anti-Trop-2 Antibody 

Preclinical efficacy has been established for TrMab-6, a new anti-Trop-2 antibody. In vitro experiments have revealed that TrMab-6 strongly induces antibody-dependent cellular cytotoxicity and complement-dependent cytotoxicity activities in mouse xenograft models of Trop-2-overexpressing CHO-K1 and breast cancer cell lines (MCF7, MDA-MB-231, and MDA-MB-468). This activity was not seen against parental CHO-K1 and MCF7/TROP2-knockout cells. In xenograft models, TrMab-6 significantly reduced tumor growth, but did not show antitumor activity in knockout xenografts, suggesting TrMab-6 could be a promising treatment option for Trop-2 expressing breast cancers. TrMab-6 can be used in flow cytometry, IHC, and Western blot analyses, therefore TrMab-6 can also be used to identify which patients may be the best responders to anti-Trop-2-targeted therapies [68].

### 6.2. Bispecific T-Cell/Trop-2 Antibody Therapy 

Liu et al. developed a novel bispecific antibody, F7AK3, that recognizes Trop-2 and CD3 [69]. Bispecific T cell engager antibodies (BiTEs) that recognize both tumor surface antigens and CD3 are a new class of immunotherapy agents, which dually engage T cells and tumor cells. The group examined F7AK3-mediated T cell activation and cytotoxicity in TNBC cell lines and primary cells in vitro and found that the cytotoxic potency was correlated with Trop-2 expression level in each of the breast cancer cell lines. F7AK3 did not bind to a Trop-2 negative breast cancer cell line or elicit any T cell cytotoxicity in the Trop-2 negative cell line, irrespective of concentration. These cytotoxic effects were confirmed in human TNBC cells. In a xenograft TNBC model, F7AK3 bispecific antibody inhibits TNBC tumor growth by recruiting T-cells and activating them within tumor tissue. These data warrant further study and clinical evaluation of F7AK3 for patients with advanced TNBC [69].

### 6.3. Anti-Trop-2 Antibody Nanoparticles

Researchers have demonstrated the Trop-2 expressing TNBC cell line, MDA-MB-231, was able to take up an antibody nanoparticle targeted to Trop-2 with encapsulated doxorubicin. Doxorubicin-loaded anti-Trop-2 antibody conjugated nanoparticles exhibited higher toxicity in the Trop-2-positive TNBC cells, compared to doxorubicin-loaded control nanoparticles without the anti-Trop-2 antibody. The goal is that an anti-Trop-2 antibody conjugated nanoparticles could be a promising carrier of doxorubicin or other chemotherapies to augment the cytotoxic effect of chemotherapy, while decreasing toxicity, similar to ADCs in the treatment of TNBC [33]. 

## 7. Conclusions 

Trop-2 overexpression has been detected across different breast cancer subtypes and is associated with increased tumor aggressiveness, metastases, and poor prognosis [1,2,3]. The emergence of Trop-2 as a promising therapeutic target with the simultaneous development of ADCs has given rise to the success of Trop-2 targeted therapies. Most importantly, these discoveries have been significantly impactful to patients with heavily pre-treated TNBC and HR+/HER2- cancers for whom good treatment options do not exist. With the FDA approval of sacituzumab govitecan in advanced and metastatic TNBC and HR+ cancers, additional Trop-2 directed ADCs such as dato-DXd, SKB264 (NCT04152499) and BAT8003 (NCT03884517) are moving to the forefront. Ongoing evaluations of Trop-2 inhibitors in combination with currently available, FDA-approved immunotherapy and targeted therapy options will be important for patients with advanced disease as well as those with early stage disease in the neoadjuvant and adjuvant setting. For now, the discovery of Trop-2 in breast cancer has given rise to a new class of novel therapeutics for our patients who have resistant disease and progressed on multiple lines of therapy, for whom novel treatment strategies are desperately needed. 

## Figures and Tables

**Table 1 cancers-14-05936-t001:** Outcomes in the ASCENT and TROPiCS-02 trials.

Clinical Trial	Clinical Outcomes	Full Population		Without Brain Metastases	
		Sacituzumab Govitecan	Chemotherapy	Sacituzumab Govitecan	Chemotherapy
Metastatic TNBC ASCENT Trial		*N* = 267	*N* = 262	*N* = 235	*N* = 233
	Median PFS, mo (95% CI)	4.8 (4.1–5.8)	1.7 (1.5–2.5)	5.6 (4.3–6.3)	1.7 (1.5–2.6)
	Median OS, mo (95% CI)	11.8 (10.5–13.8)	6.9 (5.9–7.7)	12.1 (10.7–14.0)	6.7 (5.8–7.7)
	Objective Response Rate no. of patients (%)	83 (31)	11 (4)	82 (35)	11 (5)
	Clinical Benefit Rate * no. of patients (%)	108 (40)	21 (8)	105 (45)	20 (9)
Metastatic HR+HER2- TROPiCS-02 Trial		*N* = 272	*N* = 271		
	Median PFS, mo (95% CI)	5.5 (4.2–7.0)	4.0 (3.1–4.4)		
	6-month PFS rate (95% CI)	46.1 (39.4–52.6)	30.3 (23.6–37.3)		
	9-month PFS rate (95% CI)	32.5 (25.9–39.2)	17.3 (11.5–23.2)		
	12-month PFS rate (95% CI)	21.3 (15.2–28.1)	7.1 (2.8–13.9)		
	Median OS, mo (95% CI)	14.4	11.2		
	Objective Response Rate no. of patients (%)	57 (21)	38 (14)		
	Median Duration of Response, mo (95% CI)	8.1 (6.7–9.1)	5.6 (3.8–7.9)		

* Clinical benefit was defined as a complete response, partial response, or stable disease with a duration of at least 6 months.

**Table 2 cancers-14-05936-t002:** Ongoing and completed clinical trials involving Trop-2 inhibitors as single agents in the metastatic TNBC setting.

Trial	Clinical Trials Identifier	Study Phase	Trop-2 Inhibitor	Study Design	Endpoints
SINGLE AGENTS					
Triple Negative Breast Cancer					
**ASCENT ***	NCT02574455	III	Sacituzumab govitecan	SG vs. PCT	mPFS 5.6 mo SG vs. 1.7 mo chemo; mOS 12.1 mo SG vs. 6.7 mo chemo
**SG in CNS Disease**	NCT04647916	II	Sacituzumab govitecan	SG in HER2 negative patients with at least one brain met >1.0 cm	Primary: ORR; Secondary: PFS, OS, safety & tolerability in patients with CNS disease
**TROPION** **PanTUMOR01**	NCT03401385	I	Daptopotamab deruxtecan	Dato-DXd in advanced tumors including TNBC	ORR 34% & disease control rate of 77% at 7.6 mo
**TROPION** **Breast02**	NCT05374512	III	Daptopotamab deruxtecan	Dato-DXd vs. PCT in pts who are not candidates for PD-L1 inhibitor therapy	Primary: PFS, OS Secondary: ORR, DoR, TTD, TST
**SKB264-01**	NCT04152499	I/II	SKB264-01	SKB264-01 in advanced solid tumors including TNBC	MTD/RP2D, ORR

* Studies are completed and are no longer enrolling. SG; sacituzumab govitecan, PCT; physician’s choice chemotherapy; mPFS, median progression free survival; mOS, median overall survival, ORR; objective response rate, Dato-DXd; Daptopotamab deruxtecan, DoR; duration of response, TTD; time to deterioration, TST; time to subsequent therapy, MTD; maximum tolerated dose, RP2D; recommended phase 2 dose.

**Table 3 cancers-14-05936-t003:** Ongoing and completed clinical trials involving Trop-2 inhibitors in metastatic hormone receptor-positive, HER2 negative breast cancer as single agents and in combination with immunotherapy.

Trial	Clinical Trial Identifier	Study Phase	Trop-2 Inhibitor	Study Design	Endpoints
Single Agent					
Metastatic Hormone Receptor-Positive, HER2 Negative					
**TROPiCS-02 ***	NCT03901339	III	Sacituzumab govitecan	SG vs. PCT	ORR 57% SG vs. 38% chemo; PFS 5.5 mo SG vs. 4.0 chemo; OS 14.4 mo SG vs. 11.2 chemo
**TROPION-Breast01**	NCT05104866	III	Daptopotamab deruxtecan	Dato-DXd	Primary: PFS, OS Secondary: ORR, DoR, DCR
**Combination with Immunotherapy**					
**Saci-IO HR+**	NCT04448886	II	Sacituzumab govitecan	SG +/− pembrolizumab	Primary: PFS Secondary: ORR, OS, CBR, TTP, DoR

* Studies are completed and are no longer enrolling. SG; sacituzumab govitecan, PCT; physician’s choice chemotherapy; PFS, progression free survival; OS, overall survival, ORR; objective response rate, Dato-DXd; Daptopotamab deruxtecan, DoR; duration of response, TTP; time to progression, CBR; clinical benefit rate, DCR; disease control rate, HR; hormone receptor.

**Table 4 cancers-14-05936-t004:** Ongoing and completed clinical trials involving Trop-2 inhibitors in metastatic TNBC as combination therapy.

Trial	Clinical Trial Identifier	Study Phase	Trop-2 Inhibitor	Study Design	Endpoints
Combination with Immunotherapy					
Triple Negative Breast Cancer					
**ASCENT-04**	NCT05382286	III	Sacituzumab govitecan	SG + pembrolizumab vs. PCT + pembrolizumab in PDL-1 positive	Primary: PFS Secondary: OS, ORR, DoR, TTR
**Saci-IO**	NCT04468061	II	Sacituzumab govitecan	SG + pembrolizumab in PDL-1 negative	Primary: PFS; Secondary: OS, ORR, DoR, time to ORR, TTP, CBR
**InCITe**	NCT03971409	II	Sacituzumab govitecan	SG + avelumab	Primary: BORR Secondary: ORR, CBR, mPFS, mOS, PROMs
**SG + atezolizumab**	NCT03424005	I/II	Sacituzumab govitecan	SG + atezolizumab	Primary: ORR Secondary: PFS, DCR, OS, DOR
**SG + magrolimab**	NCT04958785	II	Sacituzumab govitecan	SG + magrolimab	% of pts with DLT, AEs, PFS, ORR
**BEGONIA ***	NCT03742102	II	Daptopotamab deruxtecan	Dato-DXd + durvalumab with or without paclitaxel	ORR 66.7% dato-DXd + durva vs. 58.3% durva + chemo
**Combination with Targeted Therapies**					
**SG + PARPi**	NCT04039230	I/II	Sacituzumab govitecan	SG + talazoparib	Primary: DLTs; Secondary: time to tumor response, DoR, PFS, OS
**ASSET**	NCT05143229	I	Sacituzumab govitecan	SG + alpelisib	Primary: RP2D; Secondary: PKs, ORR

* Studies are completed and are no longer enrolling. SG; sacituzumab govitecan, PCT; physician’s choice chemotherapy; mPFS, median progression free survival; mOS, median overall survival, ORR; objective response rate, Dato-DXd; Daptopotamab deruxtecan, DoR; duration of response, TTP; time to progression, CBR; clinical benefit rate, BORR; best overall response rate, PROMs; patient related outcomes measures, DCR; disease control rate, DLT; dose-limiting toxicity, RP2D; recommended phase two dose, PK; pharmacokinetics.

**Table 5 cancers-14-05936-t005:** Ongoing and completed clinical trials involving Trop-2 inhibitors in the early stage setting as neoadjuvant and adjuvant therapy for TNBC.

EARLY STAGE SETTING					
Neoadjuvant					
**NeoSTAR ***	NCT04230109	II	Sacituzumab govitecan	SG in TNBC with at least 1 lesion > 1 cm or greater in size; combination cohort with SG + pembrolizumab followed by standard chemo	pCR at 12 weeks
**Adjuvant**					
**SASCIA**	NCT04595565	III	Sacituzumab govitecan	SG vs. PCT (observation or capecitabine) in residual disease after NACT with high risk HER2 negative disease	Primary: iDFS; Secondary: OS, distant disease-free survival, locoregional recurrence-free survival
**ASPRIA**	NCT04434040	II	Sacituzumab govitecan	SG + atezolizumab if residual disease in breast or LN or presence of ctDNA after NACT	Primary: rate of undetectable ctDNA after 6 cycles; Secondary: rate of undetectable ctDNA after 1 and 3 cycles, IDF survival rate, distant metastasis free survival rate, OS at 3 years

* Studies are completed and are no longer enrolling. SG; sacituzumab govitecan, PCT; physician’s choice chemotherapy; pCR; pathologic complete response, iDFS; invasive disease free survival, ctDNA; circulating tumor DNA, OS; overall survival.

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
