# Peer review of "Trop-2 as a Therapeutic Target in Breast Cancer"

_cancers, 2022, doi:10.3390/cancers14235936_

Round 1

Reviewer 1 Report

In this manuscript, the authors have described the Trop-2 inhibitor as a promising therapeutic target for the treatment of breast cancer. The authors firstly described the overexpression of Trop-2 in breast cancer, especially in TNBC. Next, they presented the function of Trop-2 in breast cancer. Furthermore, the authors described the current drug development by targeting Trop-2, ongoing and completed clinical trials. Overall, the authors have fully described the research status of Trop-2 targeted therapies.

Author Response

Thank you for your kind and thorough review. 

Reviewer 2 Report

Overall this is a well-written review article and covers all therapeutic options to target Trop-2 in breast cancer. 

Author Response

Thank you for your review and comments. 

Reviewer 3 Report

Interesting and comprehensive updated review of a new therapeutic target (Trop-2). The article is easy to read, is well structured and can provide current summary of the research of this new therapeutic group.

I would recommend modifying the length of Table 1 and dividing it into 2-3 sections: Metastatic Setting: Single agent (TNBC)/ (Hormone Receptor positive / Her2 negative);Metastatic Setting: combinations of Trop-2 and other agents and early breast cancer.

Author Response

Thank you for your review. We modified the length of table 1 (now called table 2-5) divided into: TNBC single agent, HR+HER2neg single agent, TNBC combination therapy, TNBC early stage therapy. Thank you for the suggestion. 

Reviewer 4 Report

The present article aims to review the work concerning Trop-2 and its targeting for breast cancer treatment. Trop-2 is detected in most cancers, its expression being higher in TNBC, although it is expressed in all types of breast cancer. Therefore, the examination of its inhibition is of extremely high interest for patients with metastatic TNBC, but it is also important for patients with hormone receptor positive breast cancer, especially those with tumors resistant to standard treatments.

The authors summarize in their review article initially the findings concerning the role of Trop-2 in breast cancer and then they present its use as a therapeutic target according to most recent literature giving emphasis to new types of treatment and the relevant clinical studies.

Author Response

Thank you for your review.